# Optimization Design of Pot Slot Structure of Tea De-enzyming and Carding Machine

Haijun Bi [1,2,*], Pengcheng Jia [1,2], Kuan Qin [3,4], Lei Yu [5], Chengmao Cao [3,4] and Yuxuan Bai [6]

1 State Key Laboratory of Tea Plant Biology and Utilization, Anhui Agricultural University, Hefei 230036, China
2 School of Tea and Food Science and Technology, Anhui Agricultural University, Hefei 230036, China
3 School of Engineering, Anhui Agricultural University, Hefei 230036, China
4 Anhui Intelligent Agricultural Machinery Equipment Engineering Laboratory, Hefei 230036, China
5 Anji Yuanfeng Tea Machinery Co., Ltd., Huzhou 313300, China
6 School of Mechanical Engineering, Hefei University of Technology, Hefei 230009, China
* Correspondence: bihaijun@ahau.edu.cn

**Abstract:** The problems of the uneven strip shape and low efficiency of tea de-enzyming and carding machines in the working process were addressed by analyzing the trajectory of tea particles and establishing a force model diagram of tea particles in the pot slot. The three-dimensional geometric model of the tea de-enzyming and carding machine was drawn using UG software, and the simulation model of tea particles was established using EDEM software. The work efficiency of the tea de-enzyming and carding machine was improved, and the rate of broken tea was reduced using the EDEM software to simulate the movement of tea particles in the pot slot under different heights of the convex bar, pot slot angle of inclination, and number of slots. The average velocity and interaction force curve of tea particles were obtained. The influence of the number of slots, the inclination angle of the slot, and the height of the convex bar on the effect of tea into strips were verified using a scheme design based on the quadratic regression orthogonal combination rotation test, and experimental research based on three factors and three levels was carried out. Design-Expert 11 software (Stat-Ease, Minneapolis, MN, USA) was used to optimize the response surface and analyze the regression model of the relevant test data. The 6CSL-800 tea de-enzyming (Anji Yuanfeng Tea Machinery Co., Ltd., Huzhou, China) and carding machine (Anji Yuanfeng Tea Machinery Co., Ltd., Huzhou, China) was used as the verification test prototype, six sets of verification tests were carried out, and the test results showed that the maximum value of the strip rate index and the minimum value of the broken tea rate index were obtained. The order of the indicators affecting the bar-type rate and broken tea rate of the de-enzyming and carding machine from high to low is as follows: the height of the convex bar, the inclination angle of the slot body, and the number of slots bodies. When the height of the convex bar was 10 mm, the inclination angle of the slot was 90°, the number of slots was 12, the bar-type rate was 89.45%, and the broken tea rate was 1.63%. The prediction results of the regression model of the bar-type rate and broken tea rate of the tea de-enzyming and carding machine were verified by employing six sets of control tests with the 6CSL-800 tea de-enzyming and carding machine as the validation test prototype. The actual values of the bar-type rate obtained from the six sets of control tests were 88.19%, 90.37%, and 87.33% (1,2,3 group), and the actual values of the broken tea rate were 1.66%, 1.69%, and 1.61% (4,5,6 group), with average values of 88.63% and 1.65%. The control test was basically consistent with the results of parameter optimization. The processed finished tea has good quality, which can provide theoretical reference for the optimization and design of tea de-enzyming and carding machines and similar tea machines in the future.

**Keywords:** tea de-enzyming and carding machine; U-slot structure; discrete element simulation; response surface method; optimization of structural parameters

## 1. Introduction

As the main traditional export agricultural product of China, tea plays a pivotal role in China's agricultural export trade [1]. Since the establishment of the People's Republic of China, the development of Chinese tea machinery has occurred for more than 70 years. With the increase in national attention to the tea machinery industry, Chinese tea processing has now basically realized mechanization and automation, and tea garden operation machinery is also developing rapidly [2]. In tea processing, the de-enzyming and carding of tea leaves are important parts of the tea processing and production process, and whether they are effective, ideal, or not is directly related to the shape and quality of the tea leaves. De-enzyming is the use of external heat to transfer heat to the tea leaves to achieve the destruction of the oxidative enzyme activity in fresh tea leaves and evaporate some of the water. This process makes the tea leaves less watery to promote the loosening of the texture of the tea leaves and facilitate the next step of kneading and forming. It also removes the green smell from tea leaves and releases aroma. In this process, the tea leaves are processed into strips, which facilitates the subsequent processing of tea leaves and the formation of important characteristics, such as color, aroma, and taste [3].

At present, the slot structure of tea de-enzyming and carding machine is mostly U-shaped. The U-shaped slot is an important part of the tea de-enzyming and carding machine. The depth and width of the U-shaped slot, the angle between the inner wall of the slot, and the horizontal surface directly affect the efficiency of the de-enzyming and carding of tea leaves [4]. At present, there are few studies on the optimization of the pot slot structure of tea de-enzyming and carding machines; they mainly focus on the shape of the pot slot and the angle between the wall and the horizontal plane. The uniformity of the tea carding was improved by Su et al. [5] by adopting a sloping U-shaped slot in the longitudinal section of the selected slot, in which one side of the slot wall was inclined from 12° to 16°, and the other side was inclined from 16° to 20°; the U-shaped slot is usually equipped with a convex strip, which can increase the resistance between the tea particles and the inner wall of the slot, which is conducive to the full turning of the tea leaves and improves the rate of forming tea leaves. Wu [6] made the pot slot body conical and equipped it with an inlet and outlet in the design of the wok groove structure of the carding machine; the width of the slot narrows from the entrance to the exit so that the volume of the tea leaves decreases because of the evaporation of water, while the volume of the U-shaped slot decreases to ensure that the space occupied by the tea leaves is evenly proportioned along the width of the slot and that the tea leaves are formed quickly and efficiently. To address the problems of good connection stability, strong thermal deformation resistance and an enhanced carding effect when working in continuous pots and slots, Huang [7] and others designed a multi-slot pot that completely eliminates the expansion suffered in the longitudinal direction and weakens the thermal deformation resistance; this structure leaves a gap between the traction tube and the pot frame on the left and right sides, and a buffer plate is placed on the front and rear slot plates to eliminate longitudinal expansion and ensure the smooth flow of tea leaves in the pot and slot. In the heating mode, according to the heating heat, there is electric heating, coal heating and oil heating [8]. Aiming at the problems of high energy consumption, slow temperature response and unequal heating, Dai et al. [9] greatly reduced energy consumption by changing the electromagnetic distance from the bottom of the pot slot and shortening the heating time of the pot slot from 20 min to 5 min to heat the tea leaves more evenly. The results of the above structural design optimization for the tea de-enzyming and carding machine are mostly in the form of patents and only focus on the shape of the slot. This design does not consider other factors of the slot that affect the efficiency of the de-enzyming and carding. Its theory and parameter optimization have not been studied, thus requiring further research.

This paper analyzes the movement trajectory and force of tea particles in the pot slot of a de-enzyming and carding machine in terms of the height of the convex bar in the pot slot, the inclination angle of the pot slot, and the number of slots. It also establishes the equation for the movement trajectory and force model of tea particles in the pot slot and

simulates the movement trajectory and distribution of tea particles in the slot by using EDEM discrete element simulation analysis software. The model is used to simulate the trajectory and distribution of tea particles in the pot slot, to optimize the structure of the pot and slot of the tea de-enzyming and carding machine and improve the efficiency of the de-enzyming and carding machine.

## 2. Materials and Methods

### 2.1. Structure and Working Principle

The 6CSL-800 gas-type tea leaf de-enzyming and carding machine is mainly composed of working parts such as a pot frame, multi-groove pot, connecting rod, motor, crank slider mechanism, gas device, gas regulating valve, gas supply pipe, and frame (Figure 1). When working, the crank slider mechanism under the driving force of the motor drives the main body of its multi-slotted pot to create reciprocal linear movement in the horizontal plane. The tea particles in the slot pot are heated by the reciprocating driving force, the friction force of the inner wall of the pot and slot on the tea particles, and the heat force. It then loses water evenly, and along the track of the pot, it is squeezed by friction and gradually carded into strips. By adjusting the frequency of the reciprocating movement of the de-enzyming and carding machine and the size of the surface temperature of the pot, the whole operation process of tea de-enzyming and carding is completed. The dried tea produced is flattened, the buds and leaves are intact, the fronds are visible, and the color is green [10]. The main working parameters of the tea de-enzyming and carding machine are shown in Table 1.

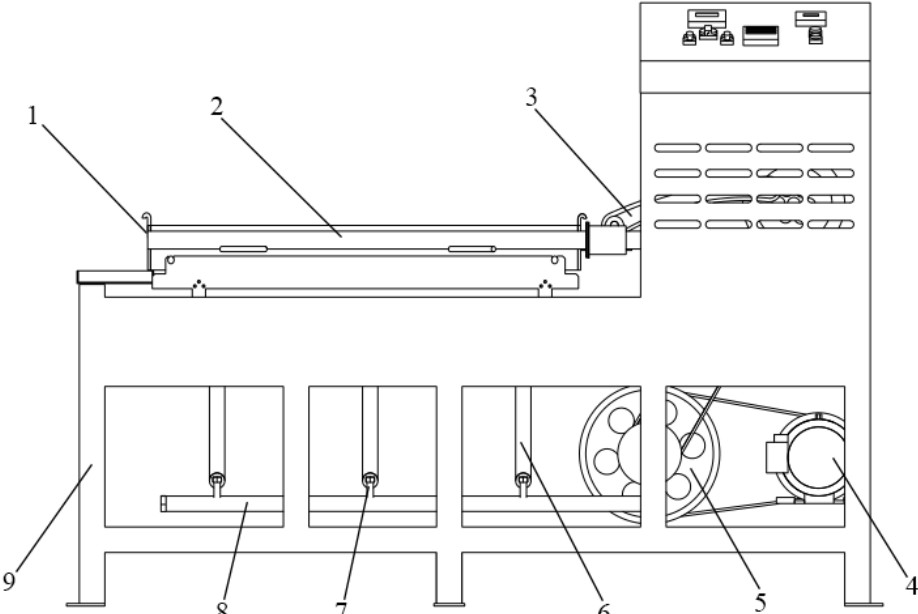

**Figure 1.** Schematic diagram of structure of tea de-enzyming and carding machine. 1. Pot rack. 2. Multi-tank pot. 3. Connecting rod. 4. Motor. 5. Crank slider mechanism. 6. Gas installations. 7. Gas control valve. 8. Gas supply pipe. 9. Rack.

### 2.2. Measuring Principle

2.2.1. Force Analysis of Tea in Pot

The movement of tea particles in the pot are as follows: (1) the reciprocating movement of tea particles along the inner wall of the pot slot occurs under the action of friction, the support force of the inner wall of the pot slot, the resistance of the convex bar in the pot slot, its own gravity, and the Coriolis force, while the throwing movement of tea particles along the pot slot is driven by the transmission mechanism; and (2) the collision movement of tea particles with the inner wall of the pot slot occurs under the action of inertial force. In the carding process, the tea particles are subjected to friction, extrusion, and rolling

when moving along the inner wall of the pot slot and the tangential extrusion when the tea particles collide with the inner wall of the pot slot so that the tea particles gradually turn into strips [11].

**Table 1.** Main working parameters of tea carding machine.

| Name | Working Parameter |
|---|---|
| Pot slot length (mm) | 1000 |
| Pot slot width (mm) | 600 |
| Motor speed (rpm) | 90–220 |
| Power of motor (kW) | 0.55 |
| Output (kg/h) | 20 |
| Form of energy | Gas |
| Dimension (mm) | $2450 \times 1100 \times 950$ |

Assuming the tea particles in motion to be cylindrical bars, with a single tea particle as the object of study and XOY as the relative coordinate system, the force analysis of the tea particles in motion within the pot is shown in Figure 2.

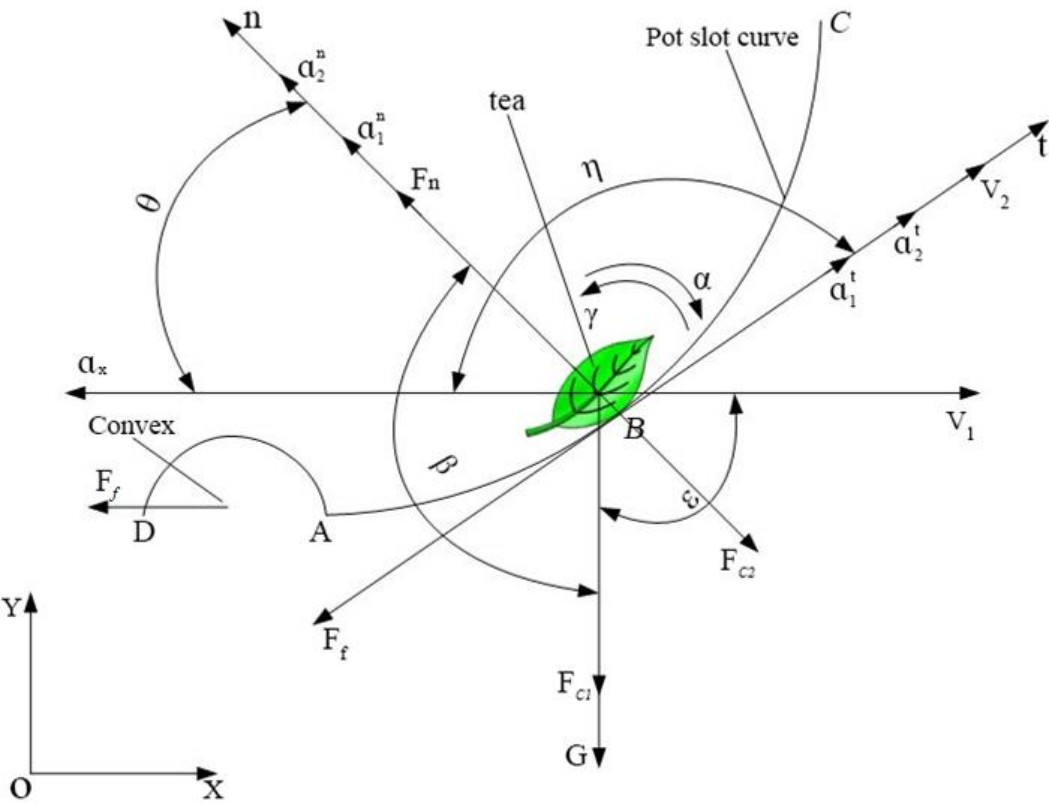

**Figure 2.** Force analysis diagram of tea particles moving in the pot.

The tea particle was studied as a mass point, the absolute coordinate system was established on the ground, and the force of the inner wall of the pot slot on the mass point of the tea particles was analysed, as shown in Figure 2. *AC* is the curve of the pot slot, *AD* represents the profile curve of the convex bar inside the pot slot, the mass center of the moving tea particle is located at point *B*, *E* is the diameter of the tea particle, *n* is the normal vector when the mass point is located on the inner surface of the pot slot, *t* is the tangential vector when the mass point is located on the inner surface of the slot, and the angular acceleration is $\alpha$.

When the tea particles are moving along the curve of the pot slot, the support force on the slot wall for the tea particles can be expressed as follows:

$$F_N = -(G + Fc_1) \cos \beta + ma_2^n + Fc_2 \tag{1}$$

The friction on the wall of the pot slot against the tea particles is as follows:

$$F_f = (G + Fc_1) \cos \varepsilon + ma_2^t = \mu F_N \tag{2}$$

in which:

$F_N$—support force of the tea particle on the wall of the pot slot, N;

$F_f$—kinetic friction between the mass point and the inner wall of the pot slot, N;

$G$—gravitational force of the tea particle (m represents the mass, g represents the gravitational acceleration, and $G$ = mg), N;

$F_{c1}$—Coriolis force of the mass point under the action of $V_1$, N;

$F_{c2}$—Coriolis force of the mass point caused by $V_2$, N;

$\beta$—angle between the gravitational force G and the normal vector $n$, (°);

$\varepsilon$—angle between the gravitational force G and the tangential vector $t$, (°);

$a_2^n$—normal component of the absolute acceleration of the mass point in the direction of the inner surface of the pot slot, mm;

$a_2^t$—tangential vector of the absolute acceleration of the mass point in the direction of the inner surface of the pot slot, mm;

$\mu$—coefficient of friction of the tea particle on the wall of the pot slot, mm.

When the tea particles are driven by the drive mechanism for the throwing motion along the pot slot, $F_N < 0$, the frictional force foes in the opposite direction to the relative average speed of the tea particles moving along the pot slot wall. The Coriolis force on the tea particles can be expressed as follows:

$$F_{c1} = -2m\gamma \times V_1 = -2ma_{c1} \tag{3}$$

$$F_{c2} = -2m\gamma \times V_2 = -2ma_{c2} \tag{4}$$

in which:

$\gamma$—angular velocity of the tea particles, rad/s;

$V_1$—speed of reciprocating motion of the pot slot during operation, mm/s;

$V_2$—relative velocity of a mass point moving against the inner wall of a pot slot, mm/s;

$a_{c1}$—Coriolis acceleration of $F_{c1}$, mm/s$^2$;

$a_{c2}$—Coriolis acceleration of $F_{c2}$, mm/s$^2$.

According to the formula for calculating the kinetic friction of an object, the resistance of the convex strip in the pot slot to the movement of tea particles along the wall of the pot slot can be expressed as follows:

$$F_f = \mu ma_x^t \tag{5}$$

in which:

$F_f$—resistance of the convex strip to the tea particles during the reciprocating movement of the pot slot, N;

$a_x^t$—tangential vector of acceleration of particles along the inner wall of the pot slot, mm.

Based on the analysis of the frictional and support forces on the tea particles, the relationship between the tangential vector $a_2^t$ and the normal vector $a_2^n$ of the absolute acceleration of the tea particles on the inner wall of the pot slot can be expressed as follows:

$$ma_2^t = \mu[(G + Fc_1) \cos \beta - ma_2^n - Fc_2] + (G + Fc_1) \cos \varepsilon \tag{6}$$

By analyzing the connection between vectors and vectors in Figure 2, the tangential vector $a_x^t$ and the normal vector $a_x^n$ of the acceleration of the tea particles along the inner wall of the pot slot can be deduced as follows:

$$a_x^t = a_x \cos \eta \tag{7}$$

$$a_2^n = a_x \cos \theta \tag{8}$$

in which:

$a_x$—the acceleration during reciprocating motion of the pot slot, mm/s$^2$;
$\eta$—the acceleration $a_1$ and the angle of the tangential vector $t$ for the motion of the pot slot, ($°$);
$\theta$—the angle between the acceleration $a_1$ and the normal vector $n$ of the motion of the pot slot, ($°$).

Considering that the acceleration of the tea leaves relative to the motion of the pot slot plus the acceleration of the inner wall of the pot slot is equal to the absolute acceleration of the tea particles, the following point acceleration summation formula can be obtained:

$$a_2^t = a_1^t + a_x^t \tag{9}$$

$$a_2^n = a_1^n + a_x^n. \tag{10}$$

in which:

$a_1^n$—the normal component of the acceleration of a mass point in the direction of movement with respect to the inner surface of the pot slot, mm;
$a_1^t$—the tangential vector quantity of acceleration of a mass point in the direction of movement with respect to the inner surface of the pot slot, mm.

Based on Equations (5)–(9), by taking the pot slot as the dynamic coordinate system, the relative acceleration component of the tea particles moving along the inner wall of the pot slot can be expressed as follows:

$$a_{1x} = a_1^t \frac{v_{2x}}{\sqrt{v_{2x}^2 + v_{2y}^2}} \tag{11}$$

$$a_{1y} = a_1^t \frac{v_{2y}}{\sqrt{v_{2x}^2 + v_{2y}^2}} \tag{12}$$

in which:

$a_{1x}$—the component of the relative tangential acceleration of tea particles moving along the inner wall of the pot slot on the X-axis, mm;
$a_{1y}$—the component of the relative tangential acceleration of the tea particles moving along the inner wall of the pot slot on the Y-axis, mm;
$v_{2x}$—the component of the relative average velocity $v_2$ of tea particles moving along the inner wall of the pot slot on the X-axis, mm;
$v_{2y}$—the component of the relative average velocity $v_2$ of tea particles moving along the inner wall of the pot slot on the Y-axis, mm.

Thus, the mass point $B$ on the relative coordinate system in the pot slot and the component between the relative velocities of tea particles can be expressed as follows:

$$\begin{cases} v_{2xB} = v_{2xB} + a_{1x}\Delta t \\ v_{2yB} = v_{2yB} + a_{1y}\Delta t \end{cases} \tag{13}$$

The trajectory of the mass point $B$ on the relative coordinate system in the pot slot can be expressed as follows:

$$\begin{cases} x_B = x_B + v_{2xB}\Delta t \\ y_B = y_B + v_{2yB}\Delta t \end{cases} \tag{14}$$

in which:

$x_{B'}$—the component on the X-axis of the mass point $B$ as it reciprocates motion and forth on the relative coordinate system of the pot slot, mm;

$y_{B'}$—the component on the Y-axis of the mass point $B$ as it reciprocates motion and forth on the relative coordinate system of the pot slot, mm;

$\Delta t$—the amount of change per unit of time in the reciprocating motion of the tea particles in the pot slot.

### 2.2.2. Force Analysis of Tea Leaves in Throwing Motion

When the angular acceleration of the tea particles is equal to zero, the angular velocity of the tea particles basically remains constant. Hence, when the air resistance is ignored, the tea particles are not affected by any external moment in the air. Therefore, when tea particles are thrown from the pot slot, only the translational motion of the tea particles needs to be studied when exploring the tea motion. The force analysis of a tea particle leaving the pot slot for the throwing motion is shown in Figure 3.

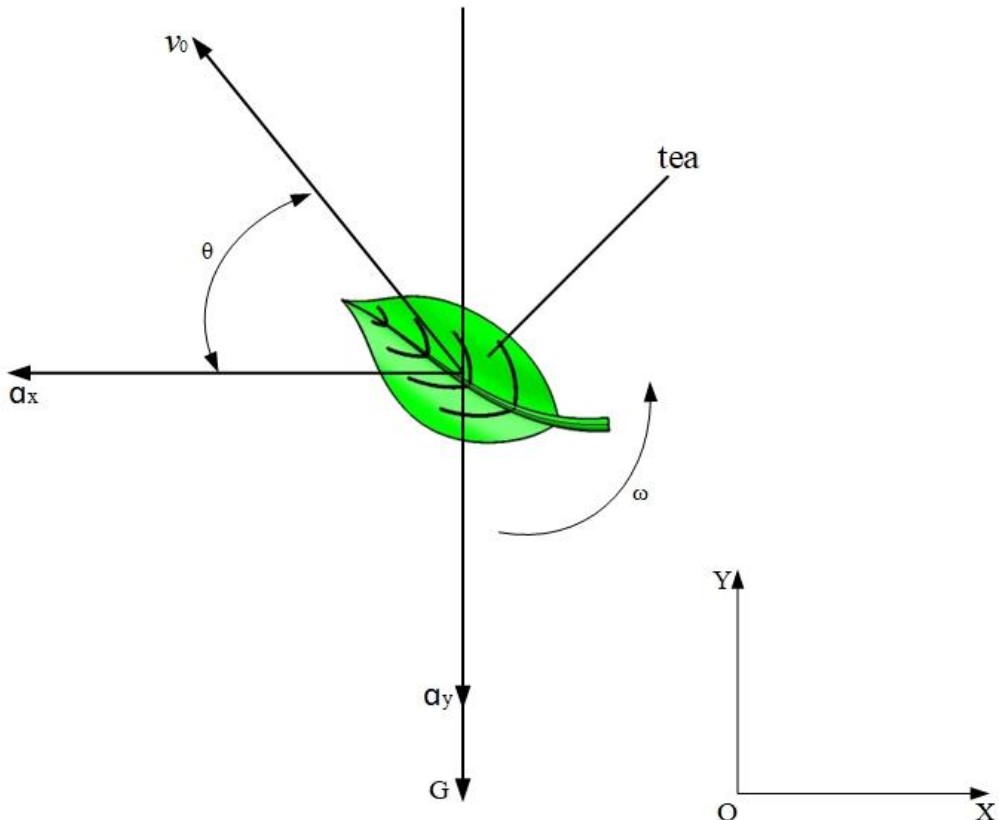

**Figure 3.** Force analysis diagram of tea particles for throwing motion.

Tea particles were studied as a single mass point, and the absolute coordinate system was established on the ground. The analysis of the throwing motion of the tea particles out of the pot slot is shown in Figure 3, in which $a_x$ is the acceleration of the mass point of the tea particle in the X-axis direction, $a_y$ is the acceleration of the tea particle mass point in the Y-axis direction, $\omega$ is the angular velocity of the tea particles leaving the pot slot for throwing motion, and $G$ is the gravity of the tea particles.

Neglecting air resistance, the equation of motion for a tea particle leaving the pot slot in a throwing motion can be expressed as follows:

$$x = v_0 t \cos\theta \tag{15}$$

$$y = v_0 t \sin\theta - \frac{1}{2}gt^2 \tag{16}$$

in which:

$x$—the component of the tea particle mass point on the X-axis, mm;
$y$—the component of the tea particle mass point on the Y-axis, mm;
$v_0$—initial velocity of the tea particle when it leaves the pot slot for the throwing movement, mm/s;
$g$—gravitational acceleration, mm/s$^2$;
$t$—the movement time of the tea particles, s;
$\theta$—the angle between the direction of the initial velocity of the center of mass of the tea particles and the direction of the X-axis, (°).

Based on the analysis of the relationship between vectors and vectors in Figure 3, the acceleration of the mass point of the tea particle in the x and y directions can be expressed as follows:

$$\begin{cases} a_x = 0 \\ a_y = g \end{cases} \tag{17}$$

The interrelationship between the velocity, acceleration, and displacement of the tea particle's mass point during the throwing motion out of the pot slot can be calculated as follows:

$$v_{0x} = v_{0x} + a_x \Delta t \tag{18}$$

$$v_{0y} = v_{0y} + a_y \Delta t \tag{19}$$

in which:

$v_{0x}{}'$—the component of the velocity $v_0{}'$ of the center of mass of the tea particle in the X-axis direction at the next moment, mm;
$v_{0y}{}'$—the component of the velocity $v_0{}'$ of the center of mass of the tea particle in the direction of the Y-axis at the next moment, mm.

$$x = x + v_{0x} \Delta t \tag{20}$$

$$y = y + v_{0y} \Delta t \tag{21}$$

in which:

$x'$—the component of the coordinates of the center of mass of the tea particles on the X-axis at the next moment, mm;
$y'$—the component of the coordinates of the center of mass of the tea particles on the Y-axis at the next moment, mm.

2.2.3. Force Analysis of the Collision Motion of Tea Leaves

Based on the force that acts between two objects in relative motion through the transfer of energy between objects when they collide, collisions can be divided into elastic and inelastic collisions. Before and after a collision, kinetic energy remains constant throughout the system, and this type of collision is called an elastic collision. The kinetic energy that is not transferred to other forms is the condition required for elastic collisions. For the whole system after the collision, impacts in which kinetic energy is converted to some extent into the internal energy of the colliding object are called inelastic impacts; the kinetic energy of the system as a whole cannot be conserved.

The collision movement between the tea particles and the inner wall of the pot slot involves collisions and deformations between rigid and flexible bodies. Friction affects the contact surface and prevents it from moving. In a short period of contact during a collision, rigid bodies are subject to frictional forces that affect the tangential direction of the contact surfaces. The resulting collision motion is mainly between the tea particles and the inner wall of the pot slot. Here, the collision between tea particles and the inner wall of the pot slot can be considered a collision between a flexible body and a rigid body on a plane with

friction. The force analysis of the collision motion between the tea particles and the inner wall of the pot slot is shown in Figure 4.

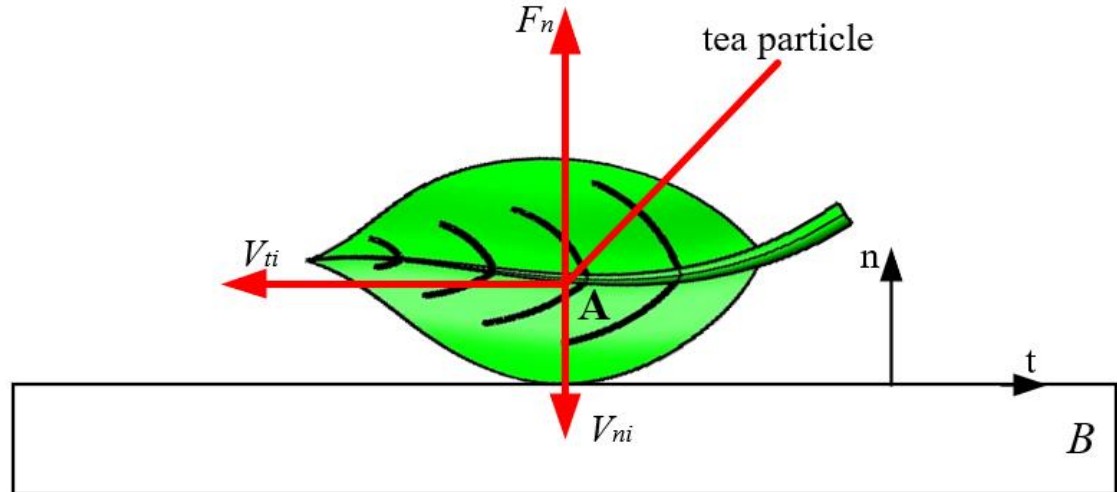

**Figure 4.** Force analysis diagram of tea particles colliding with the inner wall of the pot slot.

As shown in Figure 4, the center of mass of the tea particles is located at *A*, *B* in the wall surface of the pot slot, $\mu$ is the coefficient of friction between the tea particles and the wall of the pot slot during collision, $V_{ni}$ is the initial velocity of the mass point *A* in the vertical direction, *n* represents the unit vector in the direction of the normal, and *t* represents the unit vector in the direction of the tangent line.

During the collision of tea particles with the walls of the pot slot, the calculated normal impulse $P_n$ and tangential impulse $P_t$ are as follows:

$$P_n = \int F_n dt \qquad (22)$$

$$P_t = \int F_t dt \qquad (23)$$

in which:

$dt$—instantaneous speed, mm/s;
$F_n$—the impulse of the wall of the pot slot in the normal direction against the tea particles, N;
$F_t$—friction in the tangential direction, N.

Considering that friction is a function of the normal force between contact surfaces, the ratio of the normal force to the frictional force is equal to the coefficient of friction. Assuming that the direction of the frictional force does not change during the collision, this ratio is fixed, and the following expression can be obtained:

$$\mu = \frac{\int F_t dt}{\int F_n dt} \qquad (24)$$

Considering that the approach velocity of the two objects before the collision is proportional to the separation velocity of the two objects after the collision, the ratio of the approach velocity to the separation velocity of the two objects in the direction normal to the contact before and after the collision can be expressed as follows:

$$e = -\frac{V_{nr}}{V_{ni}} \qquad (25)$$

in which:

*e*—coefficient restitution of tea particles against the wall of the pot slot during the collision;

$V_{nr}$—normal rebound velocity of the mass center of tea particles, mm/s.

Based on Newton's second law, the normal impulse $P_n$ and the tangential impulse $P_t$ during the collision between the tea particles and the wall of the pot slot can be expressed as follows:

$$P_n = m(V_{nr} - V_{ni}) \qquad (26)$$

$$P_t = m(V_{tr} - V_{ti}) \qquad (27)$$

By combining Equations (24)–(27), the following expression can be obtained:

$$V_{ti} = V_{tr} + \mu V_{ni}(1 + e) \qquad (28)$$

in which:

$V_{tr}$—the velocity of the center of mass of the tea particles in the horizontal direction during collisional contact between the tea particles and the wall of the pot slot, mm/s;
$V_{ti}$—the initial velocity of the mass point $A$ in the horizontal direction, mm/s.

2.2.4. Force Analysis of the Interaction between Particles

When two bodies in relative motion come into contact with each other, there will be a collision force acting between them. The impact force can be expressed explicitly as a function of the embedding quantity and its rate.

In the process of tea carding, the collision between particles is the collision between two completely elastic objects. The collision between particles of tea is analyzed by referring to the linear spring-damping collision model and Hertz collision model [12]. The force analysis of the interaction between particles of tea is shown in Figure 5.

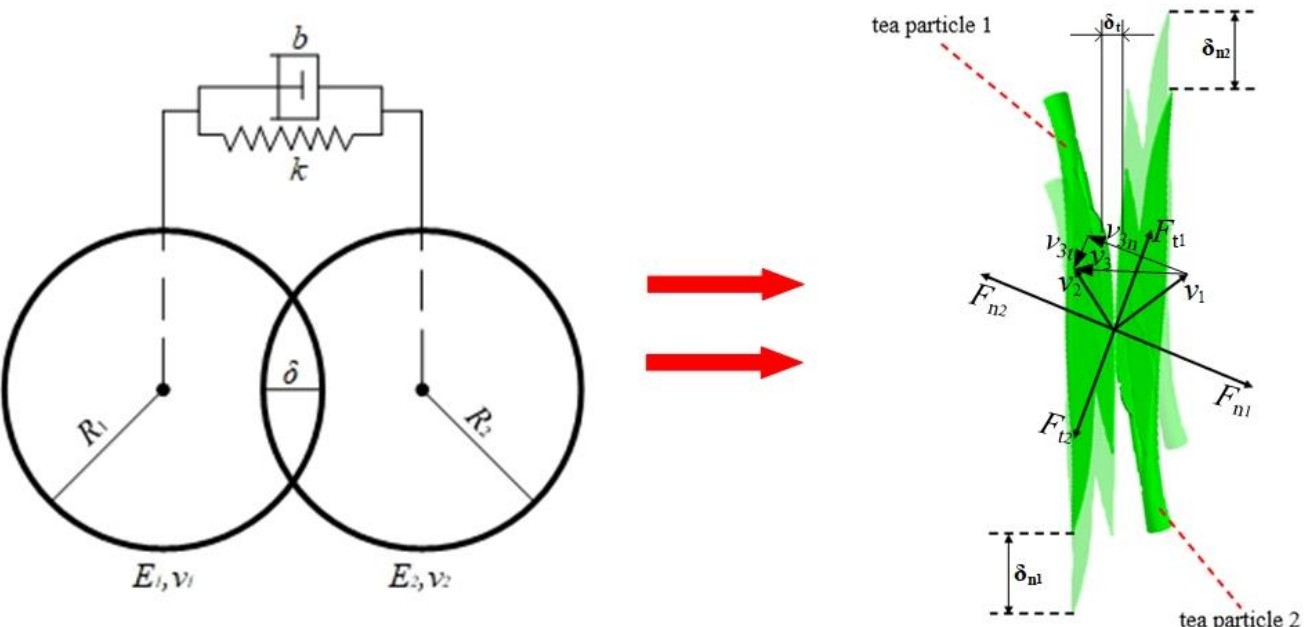

**Figure 5.** Force analysis of the interaction between particles of tea.

According to the Hertz collision model, the directional collision force $F_n$ of an object can be defined as:

$$F_n = k\delta_n \qquad (29)$$

in which:

$F_n$—the normal collision force, N;

$\delta_n$—the embedding quantity; the shape variable representing the object at the point of collision, $\delta_n > 0$, n is a constant related to the material and geometric characteristics of the colliding object;

$k$—the coefficient of elasticity; $k = (4/3)ER^{1/2}$, $R$ is the comprehensive radius of curvature, mm.

According to the above equation:

$$\frac{1}{E} = \frac{\left(1 - v_1^2\right)}{E_1} + \frac{\left(1 - v_2^2\right)}{E_2} \tag{30}$$

in which:

$E$—synthesizes Young's modulus;
$E_1$, $E_2$—Young's modulus of the colliding object;
$v_1$, $v_2$—Poisson's ratio of the colliding object.

where the expression of normal force $F_{n1}$ is:

$$F_{n1} = \frac{4}{3} E_1 \sqrt{R_1 \delta_{n1}} \tag{31}$$

in which:

$R_1$—the radius of curvature of the colliding object, mm;
$\delta_{n1}$—the normal overlap quantity, mm.

where the expression of normal damping force $F_{n2}$ is:

$$F_{n2} = -2\sqrt{\frac{5}{6}} \frac{\ln e}{\ln^2 e + \pi^2} \sqrt{S_{n1} m_1} v_{3n} \tag{32}$$

in which:

$e$—the coefficient of recovery;
$v_{3n}$—the normal component of the relative velocity, mm/s;
$m_1$—the equivalent mass, kg;
$S_{n1}$—the normal stiffness, N/mm.

The expression of the tangential force $F_{t2}$ is:

$$F_{t2} = -S_{t2} \delta_{t2} \tag{33}$$

in which:

$S_{t2}$—the tangential stiffness, N/m;
$\delta_{t2}$—the tangential overlap, m.

where the expression of tangential damping force $F_{t1}$ is:

$$F_{t1} = -2\sqrt{\frac{5}{6}} \frac{\ln e}{\sqrt{\ln^2 e + \pi^2}} \sqrt{S_{t1} m_1} v_{3t} \tag{34}$$

in which:

$v_{3t}$—the tangential component of the relative velocity, m/s.

## 3. Simulation Test

### 3.1. Modeling of the De-enzyming and Carding Machine

Further simulations and calculations were carried out using UG to model the entire structure of the tea de-enzyming and carding machine in three dimensions. The 3D model of the whole machine was simplified appropriately. The frame, the gas appliances, and the motor, which are not in direct contact with the tea particles in the pot slot, were removed.

The file was saved in .stp format and imported into the EDEM software for motion process simulation. The simplified model is shown in Figure 6.

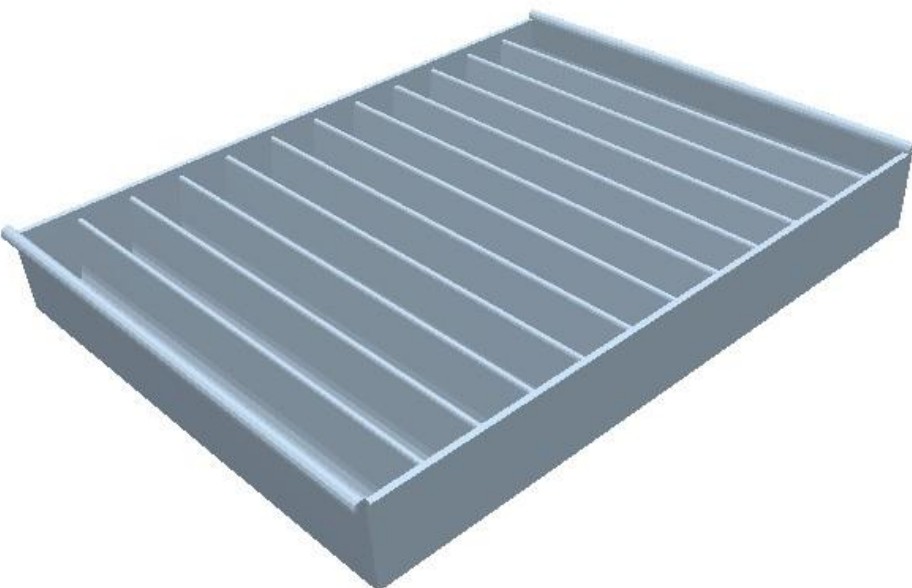

**Figure 6.** Simulation model imported into EDEM.

*3.2. Tea Particle Modelling and Simulation Parameter Setting*

Considering that the simulation process mainly facilitates the observation of the trajectory of the tea particles in the pot slot and the full turning of the tea particles in the pot slot, tea particles with different shapes will increase the simulation running time. In practice, the axial length of tea is larger than the radial length of tea. Therefore, a tea particle cannot be assumed to be a spherical particle. Accordingly, the tea was simplified to grow into strip particles, as shown in Figure 7. Except for the tea particles, the material property of the de-enzyming and carding machine pot body was set to 316 stainless steel. Material properties include Poisson's ratio, density, the shear modulus, the coefficient of static friction between particles, the coefficient of rolling friction between particles and other parameters of the particle model. The other parameters also include Poisson's ratio, density, the shear modulus of the geometric model material, the static friction factor of the particle-geometric model, and the rolling friction factor of the particle-geometric model [13]. Access to relevant literature [14], the material properties of the pot were set as follows: Poisson's ratio was set to 0.25, density was set to 7800 kg·m$^{-3}$, and the shear modulus was set to $7 \times 10^7$ Pa. The specific simulation parameters are shown in Table 2.

*3.3. Motion Simulation Process*

Virtual simulation tests were carried out in EDEM software, including the analysis of the main causes of problems, such as low bar-type rate, bursting, excessive water, red stems, and red leaves caused during the de-enzyming and carding process, modification of pot slots by different convex bar heights, number of slots and baffle angles. The tests aimed to study the effect of these factors on the effect of de-enzyming and carding. The optimization of the design of the tea de-enzyming and carding machine and its key components will provide useful information for subsequent work.

3.3.1. Creating a Particle Factory

The no-slip contact Hertz–Mindlin model (no slip) was set as the contact type for tea particles. The total number of tea particles produced was set to 5000, producing a tea particle count of 2500 per s, the start time was set to $1 \times 10^{-1}$–12 s, and the time interval

of 0.01 s was set for EDEM recording data during simulation. The settings are shown in Figure 8 [15].

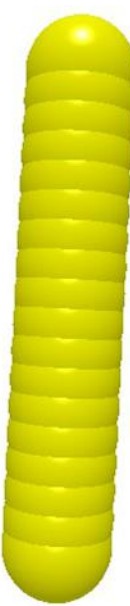

**Figure 7.** Tea particle simulation model.

**Table 2.** Parameters used in simulation setting.

| Materials | Parameters | Values |
|---|---|---|
| Tea particle | Poisson's ratio | 0.4 |
| | Density/(kg·m$^{-3}$) | 562.4 |
| | Shear modulus/Pa | $1 \times 10^7$ |
| 316 stainless steel | Poisson's ratio | 0.3 |
| | Density/(kg·m$^{-3}$) | 78.00 |
| | Shear modulus/Pa | $7 \times 10^7$ |
| Particle-particle | Coefficient of restitution | 0.37 |
| | Coefficient of static friction | 0.8 |
| | Coefficient of rolling friction | 0.05 |
| Particle-316 stainless steel | Coefficient of restitution | 0.4 |
| | Coefficient of static friction | 0.65 |
| | Coefficient of rolling friction | 0.02 |

3.3.2. Set Simulation Time and Grid Size

The number of simulation steps, time, data saving interval, and grid size were set in the EDEM solver [16] to ensure the smooth running of the simulation process. The fixed time step was set to $5.2 \times 10^{-5}$ s (20% of the Rayleigh time step), and the total simulation time was 10 s (0–2 s for the filling process of the particles and 2–10 s for the machine to perform the reciprocating motion until the end of the carding process); the data were saved at intervals of 0.01 s, and the cell size of the simulation grid was set to three times the average radius of the particles for accurate processing of subsequent data; finally, the simulation was started, and the EDEM simulation model of the machine's operating process is shown in Figure 9 [17,18].

**Figure 8.** Particle factory settings.

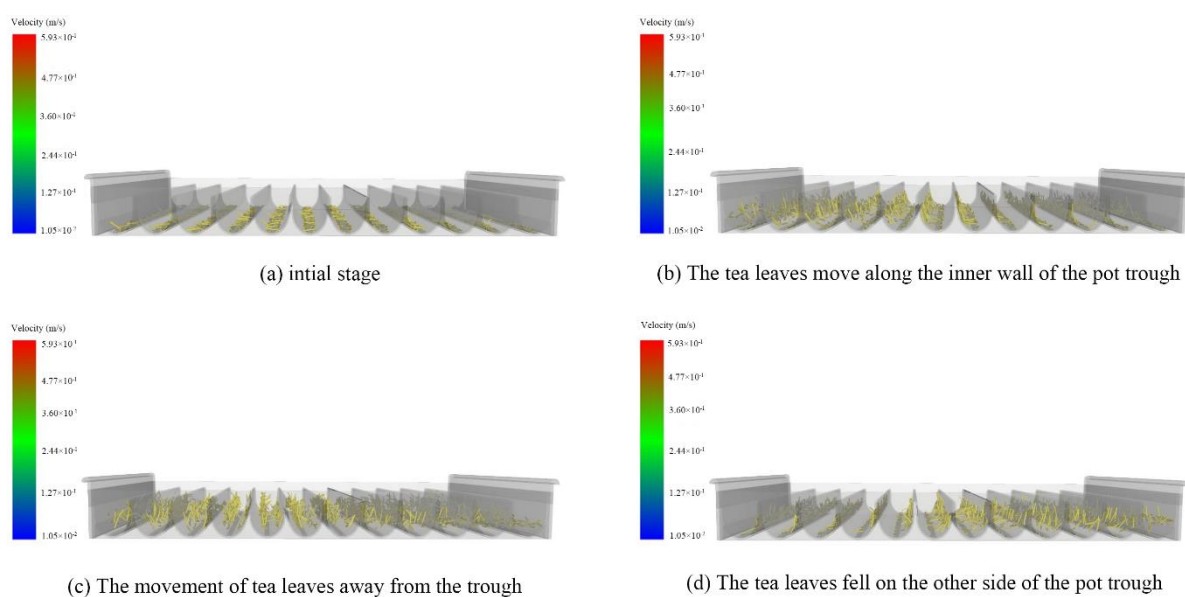

Figure 9. Simulation model of de-enzyming and carding machine in EDEM.

## 4. Analysis of Simulation Results

The working principle of the tea de-enzyming and carding machine can be derived. The effect of tea de-enzyming and carding can be determined based on whether tea particles can be turned evenly in the pot slot. The height of the convex bar, the angle of inclination of the slot, and the number of slots were used as influencing factors for the simulation of the working process of the tea de-enzyming and carding machine in EDEM. Based on the analysis of the average velocity of tea particles in the pot body during movement, as well as the curve of the interaction force with time, factors such as the height of the convex bars of the pot slot, the angle of inclination of the slot, and the number of slots affect the de-enzyming and carding of tea leaves.

### 4.1. Influence of Different Convex Bar Heights on the Effect of Tea De-enzyming and Carding

In both theory and practice, the convex bars in the pot slot play a vital role in the de-enzyming and carding of tea leaves, as shown in Figure 10; the convex bar structure facilitates the carding of the tea leaves in the carding process and promotes full turning and carding [19]. As shown in Figure 10a, changing the height of the convex bar in the pot slot has no significant effect on the average velocity of the tea particles but affects the interaction forces between the tea particles. Figure 10b shows that when the height of the convex in the pot slot is 5, 10, or 15 mm, the mean values of the interaction forces present between the tea particles are 0.35, 0.76, and 0.87 N. The effect of the height of the convex strip on the force applied to the tea particles is divided into two stages. At 0–2 s, the tea particles are driven by the mechanism and start to move along the inner wall of the pot slot; in the interaction, forces between tea particles gradually increase. The set-up was stabilized after 2 s, and the height of the convex bar remarkably influenced the force on the tea particles. When the working speed of the de-enzyming and carding machine was constant, at convex bar heights of 10 and 15 mm, a remarkably stronger interaction was observed between tea particles with the increase in time. Moreover, collision and compression forces between particles increased, thus allowing the tea leaves in the pot slot to be fully turned over, which is conducive to turning tea into strips. When the height of the convex bar in the slot was 5 mm, with the increase in time, as the resistance of the tea leaves to the inner wall of the pot slot decreased, a very small interaction force was observed between the tea particles. Moreover, the tea leaves did not turn over sufficiently in the pot slot, resulting in an uneven force on tea particles, and the strip formation rate was low.

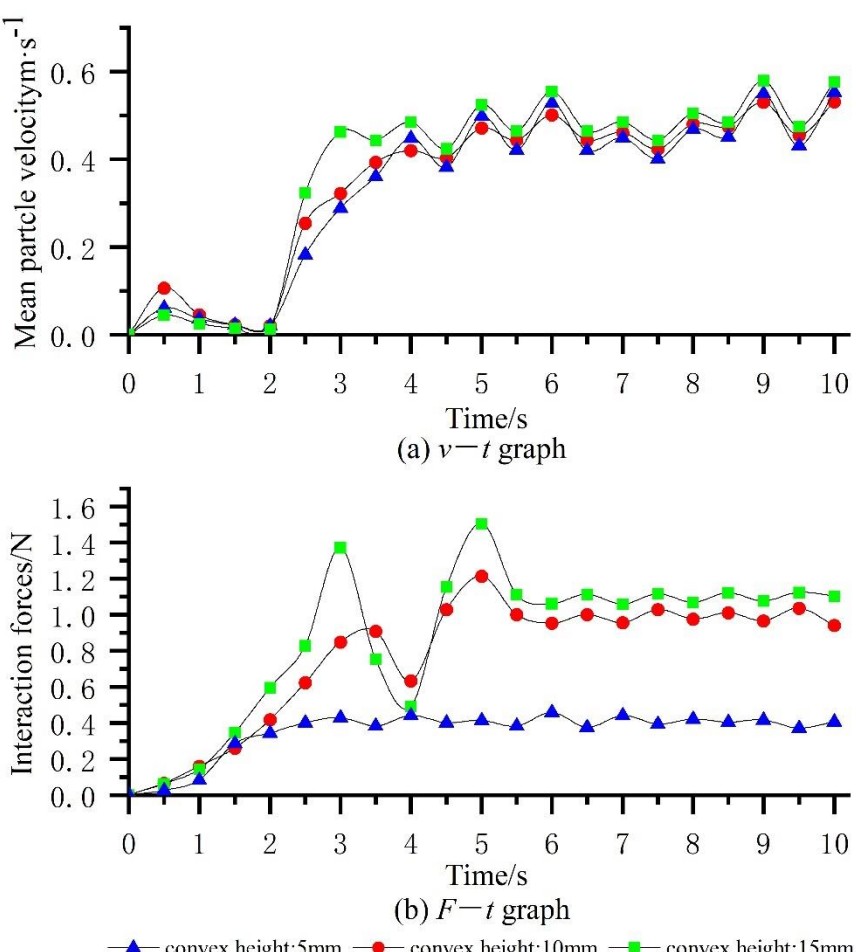

**Figure 10.** Average velocity and interaction force curves of tea particles at different heights of convex bar.

*4.2. Influence of Different Pot Slot Inclination Angles on the Effect of Tea De-enzyming and Carding*

When working on the de-enzyming and carding machine, the angle of inclination of the pot slot can directly affect the axial displacement of the tea particles in the slot and the contact time between the tea particles and the inner wall of the pot slot, thus affecting the bar-type rate of tea particle; therefore, the efficiency of the tea de-enzyming and carding machine at different pot slot inclination angles needs to be studied [20]. When the working speed of the de-enzyming and carding machine was kept constant, at a very small angle of inclination of the pot slot, the tea particles moving along the inner wall of the pot slot had a small axial displacement, which predisposed the machine to the accumulation of material in the pot slot [21]. Figure 11a shows that when all parameters were equal, the increase in the inclination angle of the pot slot resulted in a high axial displacement of the tea particles, and the average velocity of the tea particles was remarkably increased. Moreover, the tea particles gradually became larger because of the mutual friction between particles, collision extrusion, and internal friction, and the tea leaves were gradually carded into strips. The inclination angle of the different pot slots affected the average speed of the tea particles. At 0–5 s, the average speed of the tea particles was gradually increased by the driving force. Levelling off occurred after 5 s. When the pot slot was inclined at an angle of 80°, the average velocity of tea particles was relatively small and stabilized at around 0.44 mm/s, causing the tea particles to remain in the pot slot for too long, as they did not leave the pot slot for the throwing motion. This caused the tea leaves to "pop" and resulted in "red stems and leaves". When the pot slot was inclined at an angle of 100°, the average rate of the tea particles was relatively large and reached 0.36 mm/s, and the

axial travel distance of the tea particles increased, causing the tea particles to stay in the pot slot for a shorter period of time. This condition easily caused "de-enzyming is not ripe" and "too much water". When the pot slot was inclined at an angle of 90°, the average rate of tea particles was medium and largely stable at 0.30 mm/s, with good throwing and axial movement of the tea particles in the pot slot. As shown in Figure 11b, different pot slot inclination angles remarkably affect the interaction forces between tea particles. At 0–5 s, the interaction forces between tea particles increased, and a steady state was reached after 5 s. When the pot slot was inclined at an angle of 80°, low average squeezing pressure was observed between the tea particles. The interaction force between the tea particles remained stable at around 0.23 N. The extrusion and friction between the inner wall of the pot slot and the tea particles are very small. The axial travel distance of tea particles was small. Under prolonged contact in the pot slot, the efficiency of tea de-enzyming and carding decreased. When the pot slot was inclined at an angle of 100°, the average squeezing pressure between tea particles increased. The interaction force between the tea particles remained stable at around 0.62 N, and the squeezing pressure between the tea particles and the inner wall of the pot slot remarkably increased. The excessive distance regarding the axial movement of the tea particles keeps them in the pot slot for a short time, not meeting the required process requirements for de-enzyming. When the pot slot was inclined at an angle of 90°, the average value of the interaction force between the tea particles was 0.35 N. The pressure and collision between the inner wall of the pot slot and the tea particles allowed the tea particles to complete the de-enzyming and carding process in time, thus meeting the process requirements for the de-enzyming and carding process.

### 4.3. Influence of Different Numbers of Slots on the Effect of Tea De-enzyming and Carding

The number of slots affects the effectiveness of the de-enzyming and carding of tea leaves. As shown in Figure 12a, at a constant working speed of the de-enzyming and carding machine, when 10, 12, and 15 slots were used, the average velocities of tea particles were 0.30, 0.36, and 0.40 mm/s, respectively. At 0–2 s, the average velocity curve of tea particles shows a rising and then decreasing trend. The average velocity curve of tea particles at 2–5 s showed a rising trend, reached its maximum value at 7 s, and then leveled off. The average speed of the 10 slot pots is very small. When tea particles stayed in the pot slot for a long time, de-enzyming overkill occurred, resulting in the low quality of the finished tea. The average speed of 15 slot pots was too large, with a short residence time of tea leaves in the pot slot, resulting in too short a de-enzyming time and making it easy to produce the "de-enzyming is not ripe" phenomenon. As shown in Figure 12b, when 10, 12, and 15 slots were used at a constant working speed of the de-enzyming and carding machine, the mean values of the interaction forces between tea particles were 0.24, 0.30, and 0.53 N. When the number of slots was 10, low interaction forces were observed between tea particles, less collision and contact action were observed between particles, and tea particles did not turn over sufficiently in the pot slot. When the number of slots was 15, excessive interaction forces were observed between tea particles, and intense contact and collisions were observed between particles. Consequently, the rate of broken tea increased, thus affecting the overall quality of the finished tea. Therefore, selecting the right number of slots for de-enzyming and carding tea leaves can remarkably increase efficiency and thus improve the overall quality of the finished tea.

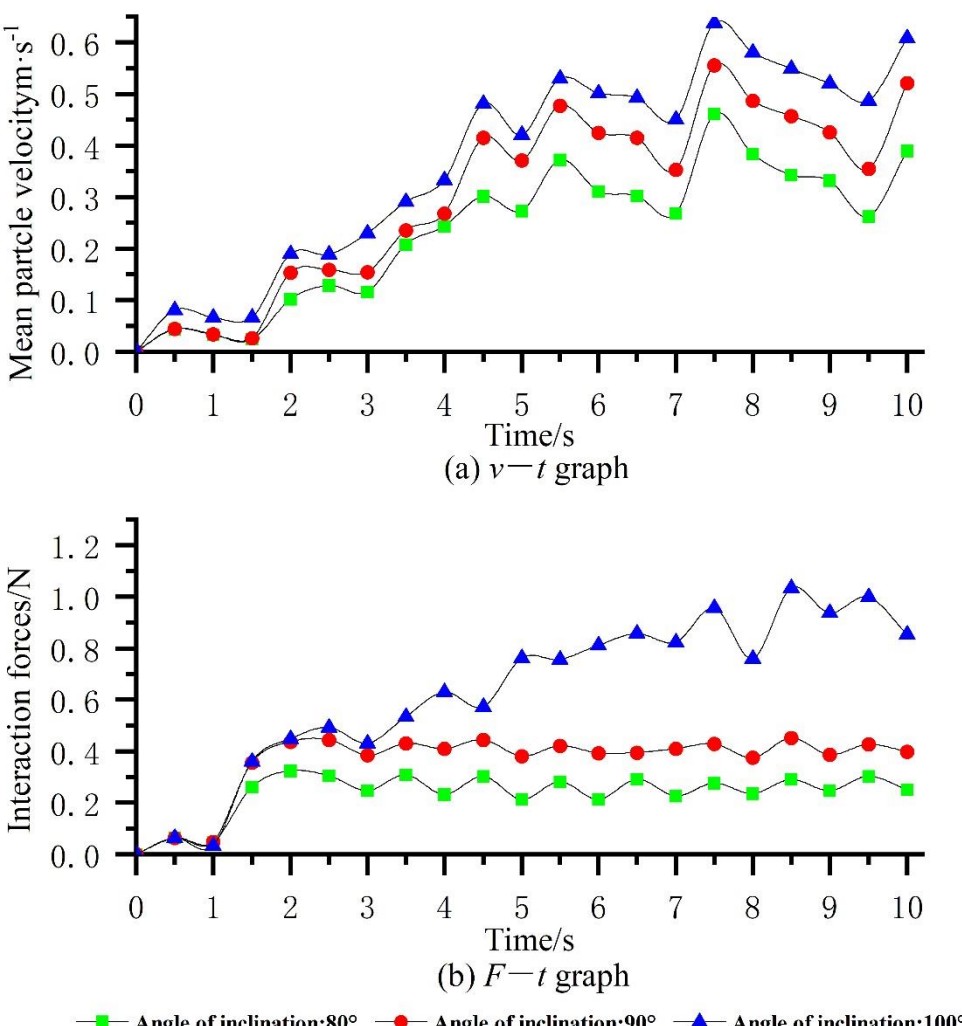

**Figure 11.** Average velocity and interaction force curves of tea particles at different pot slot angles of inclination.

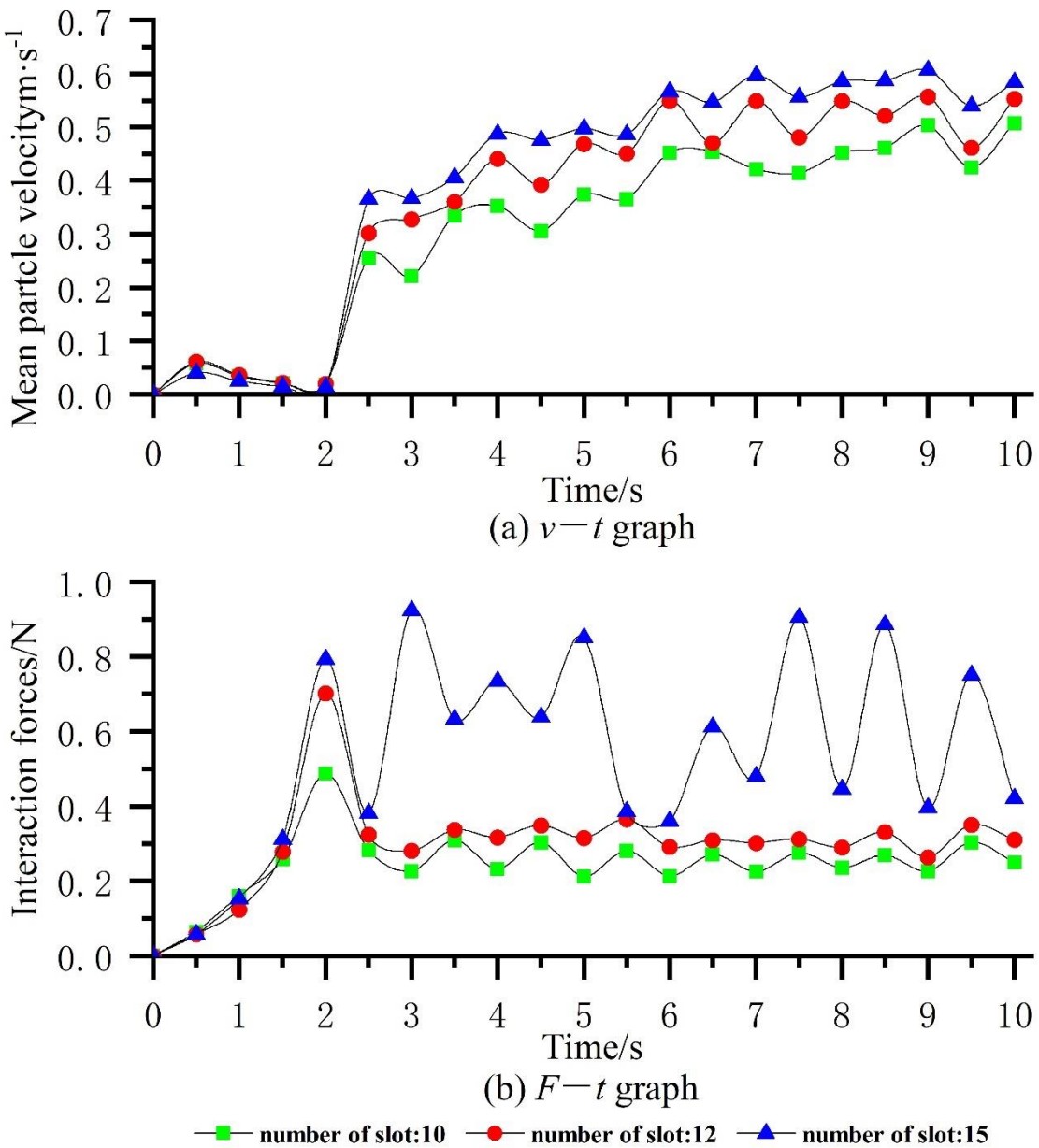

**Figure 12.** Average velocity and interaction force curves of tea particles at different numbers of pot slots.

## 5. Results Analysis and Test

### 5.1. Experimental Design

The response surface method is an experimental analysis method used to analyze the relationship between multiple influences and their response values in the presence of interactions. A polynomial regression model [22–24] was used to further verify the influence of the number of slots, the angle of inclination of the pot slot, and the height of the convex bars on the effect of the de-enzyming and carding of tea leaves when the constant pot slot temperature, the leaf feed, and the hourly output were unchanged through single-factor experiments and response surface analysis with selected factors (e.g., height of the convex bars (A), slot inclination angle (B) and the number of slots (C)), the bar-type rate and broken tea rate were used as the evaluation indexes of the optimized experimental design. The scheme design was carried out according to the quadratic regression orthogonal combination rotation test. An experimental study based on three factors and three levels was carried out, and the test factors and levels are shown in Table 3 [25,26].

**Table 3.** Factors and levels of test.

| Code | Convex Height | Pot Slot Angle of Inclination | Number of Slots |
|---|---|---|---|
| −1 | 5 | 80 | 10 |
| 0 | 10 | 90 | 12 |
| 1 | 15 | 100 | 15 |

*5.2. Test Results and Analysis of Variance*

The test factors were analyzed using Design-Expert 11 software(Stat-Ease; Minneapolis; The United States). The results of this calculation are shown in Table 4. The results obtained from Table 4 were subjected to model ANOVA and significance tests. The results obtained are shown in Table 5. The credibility analysis of the model is shown in Table 6 [27].

**Table 4.** Calculation results.

| Test Number | Test Influencing Factors | | | Evaluation Index | |
|---|---|---|---|---|---|
| | Convex Height/(mm) | Pot Slot Angle of Inclination/(°) | Number of Slots/(pcs) | Bar-Type Rate/% | Broken Tea Leaves Rate/% |
| 1 | 10 | 100 | 10 | 85.26 | 2.28 |
| 2 | 15 | 100 | 12 | 87.52 | 1.93 |
| 3 | 10 | 90 | 12 | 89.45 | 1.63 |
| 4 | 5 | 90 | 15 | 84.67 | 2.14 |
| 5 | 10 | 90 | 12 | 89.29 | 1.54 |
| 6 | 15 | 80 | 12 | 86.28 | 1.86 |
| 7 | 10 | 90 | 12 | 89.19 | 1.61 |
| 8 | 10 | 90 | 12 | 89.37 | 1.49 |
| 9 | 10 | 100 | 15 | 85.12 | 2.32 |
| 10 | 10 | 90 | 12 | 88.97 | 1.69 |
| 11 | 5 | 90 | 10 | 84.33 | 2.36 |
| 12 | 15 | 90 | 15 | 86.68 | 1.96 |
| 13 | 5 | 100 | 12 | 83.86 | 2.47 |
| 14 | 5 | 80 | 12 | 84.21 | 2.16 |
| 15 | 10 | 80 | 15 | 83.76 | 1.89 |
| 16 | 10 | 80 | 10 | 84.78 | 2.24 |
| 17 | 15 | 90 | 10 | 87.55 | 1.82 |

Through the analysis of the variance of Table 5, the quadratic polynomial between the factor and the response parameter was fitted. Finally, the regression equations $Y_1$ for the rate of tea formation and the regression equation $Y_2$ for the rate of broken tea were derived, as expressed in Equations (29) and (30) [25].

$$Y_1 = 89.30 + 1.34A + 0.3651B - 0.2112C + 0.3975AB - 0.3088AC + 0.2390BC - 1.35A^2 - 2.43B^2 - 2.14C^2 \quad (35)$$

$$Y_2 = 1.57 - 0.1859A + 0.1160B - 0.0487C - 0.0600AB + 0.0912AC + 0.0978BC + 0.2022A^2 + 0.3127B^2 + 0.2995C^2 \quad (36)$$

As shown in Table 6, the coefficient of determination $R^2$ for the bar-type rate ($Y_1$) is 0.9964, the corrected coefficient of determination $R^2_{adj}$ is 0.9917, and the coefficient of variation CV is 35.51%. The regression relationship can explain 99.17% of the variation of the model. The overall probability of variation in the model is within 0.0001, indicating that the model can accurately reflect the variation of the bar-type rate with the height of the convex bar, the angle of inclination of the slot, and the number of slots. The overall *F*-value test of the model revealed that the probability of variation in the misfit of the model was not significant, the model fits well, and the influencing factors and the bar-type rate have a non-linear relationship. As shown in Table 5, the magnitude of the three factors influencing

factor values is, in order, as follows: the height of the convex bar is 365.98, the pot slot angle of inclination is 27.21, the number of slots is 9.28, and the proportions of relative values are 90.93%, 6.76%, and 2.30%. The height of the convex strip plays a decisive role in the influence of the bar-type rate. The angle of inclination of the slot and the number of slots affect the bar-type rate by 7% and 2%, respectively. As shown in Table 5, considering that a higher *F*-value results in a more significant effect of the factor, the order of influence of the interaction between the factors on the significance of the bar-type rate is AC < AB. As shown in Table 6, the coefficient of determination $R^2$ for the broken tea leaf rate ($Y_2$) is 0.9796, $R^2_{adj}$ is 0.9534, and CV is 3.41. The regression relationship could explain 95.34% of the variation of the model. The overall variation probability of the model is less than or equal to 0.0001, indicating that the model can accurately reflect the relationship of the rate of broken tea leaves with the height of the convex bars, the angle of inclination of the slot, and the number of slots. The model was tested by the *F*-value, and the results show that the variation probability of the misfit degree was not significant, indicating that the fitting degree was high, and a non-linear correlation was observed between the influencing factors and the rate of broken tea. As shown in Table 5, the magnitude of the three factors influencing factor values is, in order, as follows: the height of convex bar is 60.45, the sloped angle of the pot slots is 23.55, the number of slots is 4.24, and the proportions of relative value are 68.51%, 26.69%, and 4.80%, respectively. The influence of the convex bar height on the percentage of broken tea accounted for more than half, while the pot slot is inclined at about a quarter of the angle. The effect of the number of slots on the rate of broken tea is only close to 5%. As shown in Table 5, considering that a larger *F*-value results in a more significant role of the factor, the order of influence of the interaction between the factors on the significance of the broken tea leaves rate was AC > AB.

**Table 5.** Analysis of variance.

| Evaluation Index | Source | Sum of Squares | df | Mean Square | F-Value | p-Value |
|---|---|---|---|---|---|---|
| | Model | 74.19 | 9 | 8.24 | 214.43 | <0.0001 |
| | A | 14.07 | 1 | 14.07 | 365.98 | <0.0001 |
| | B | 1.04 | 1 | 1.04 | 27.21 | 0.0012 |
| | C | 0.35 | 1 | 0.35 | 9.28 | 0.0187 |
| | AB | 0.63 | 1 | 0.63 | 16.44 | 0.0048 |
| | AC | 0.38 | 1 | 0.38 | 10.12 | 0.0155 |
| Bar-type rate $Y_1$ | BC | 0.23 | 1 | 0.23 | 6.06 | 0.0433 |
| | $A^2$ | 7.72 | 1 | 7.72 | 200.94 | <0.0001 |
| | $B^2$ | 24.90 | 1 | 24.90 | 647.79 | <0.0001 |
| | $C^2$ | 17.32 | 1 | 17.32 | 450.67 | <0.0001 |
| | Residual | 0.26 | 7 | 0.03 | | |
| | Lack of fit | 0.13 | 3 | 0.04 | 1.27 | 0.3980 |
| | Pure error | 0.14 | 4 | 0.03 | | |
| | Cor Total | 74.46 | 16 | | | |
| | Model | 1.50 | 9 | 0.17 | 37.37 | <0.0001 |
| | A | 0.27 | 1 | 0.27 | 60.45 | 0.0001 |
| | B | 0.10 | 1 | 0.11 | 23.55 | 0.0018 |
| | C | 0.01 | 1 | 0.02 | 4.24 | 0.0785 |
| | AB | 0.01 | 1 | 0.01 | 3.21 | 0.1163 |
| Broken tea leaves rate $Y_2$ | AC | 0.03 | 1 | 0.03 | 7.56 | 0.0285 |
| | BC | 0.03 | 1 | 0.04 | 8.70 | 0.0214 |
| | $A^2$ | 0.16 | 1 | 0.17 | 37.65 | 0.0005 |
| | $B^2$ | 0.41 | 1 | 0.41 | 91.83 | <0.0001 |
| | $C^2$ | 0.34 | 1 | 0.34 | 76.00 | <0.0001 |
| | Residual | 0.03 | 7 | 0.004 | | |
| | Lack of fit | 0.01 | 3 | 0.002 | 0.38 | 0.78 |
| | Pure error | 0.02 | 4 | 0.01 | | |
| | Cor Total | 1.54 | 16 | 0.17 | | |

**Table 6.** Reliability analysis of the model.

| Evaluation Index | Parameters | Values |
|---|---|---|
| Bar-type rate $Y_1$ | Coefficient of determination, $R^2$ | 0.9964 |
| | Adjusted coefficient of determination, $R^2_{adj}$ | 0.9917 |
| | Standard deviation, Std. Dev. | 0.1961 |
| | Coefficient of variation, CV | 35.51 |
| Broken tea leaf rate $Y_2$ | Coefficient of determination, $R^2$ | 0.9796 |
| | Adjusted coefficient of determination, $R^2_{adj}$ | 0.9534 |
| | Standard deviation, Std. Dev. | 0.0670 |
| | Coefficient of variation, CV | 3.41 |

As shown in Table 5, the constructed bar-type rate model has significant regression ($p < 0.0001$). Its coefficient of determination $R^2$ reached 0.9964 and approached 1, indicating that the model and simulation data are highly feasible, and it can accurately predict the bar-type rate. The constructed broken tea leaf rate model has significant regression ($p < 0.0001$). Its coefficient of determination $R^2$ reached 0.9796 and approached 1, indicating that the model and simulation data are highly feasible, and it can accurately predict the broken tea leaf rate. The calculated and predicted values of the model for bar-type rate and broken tea leaf rate are shown in Figure 13 [25].

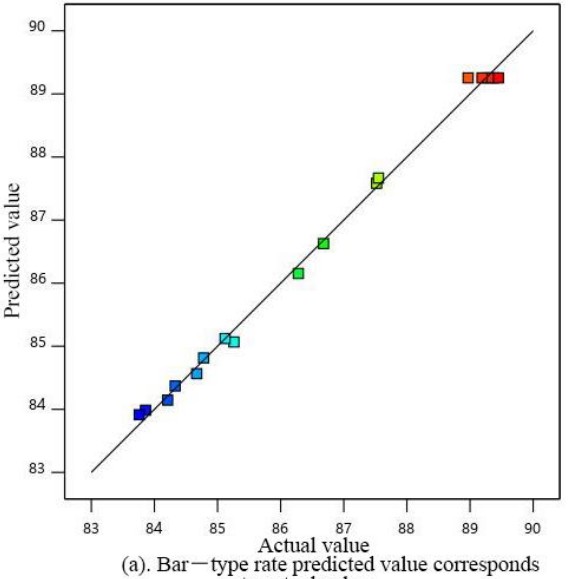 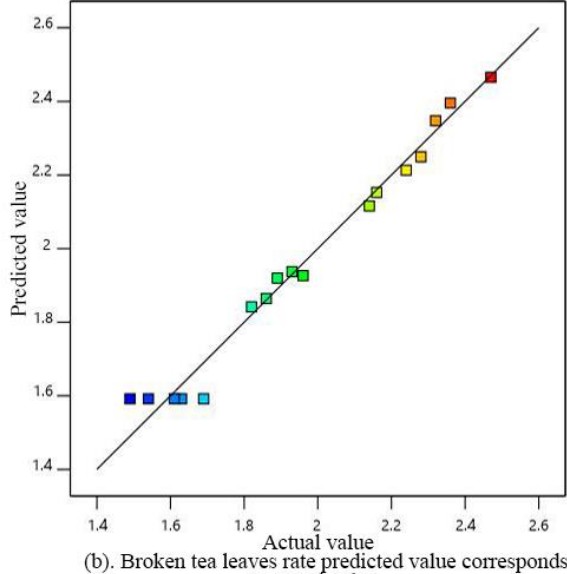

**Figure 13.** Analysis of calculated and predicted values of bar-type rate and broken tea leaf rate.

A residual plot of the fitted residuals for the rate of bar-type and broken tea is shown in Figure 14. This plot aims to show if the observed data are outliers; if the regression fit is low, then the larger the residual, the further away each point is from the oblique line [27]. As shown in Figure 14, the points are on each side of the diagonal line and close to it, indicating the goodness of fit of the test.

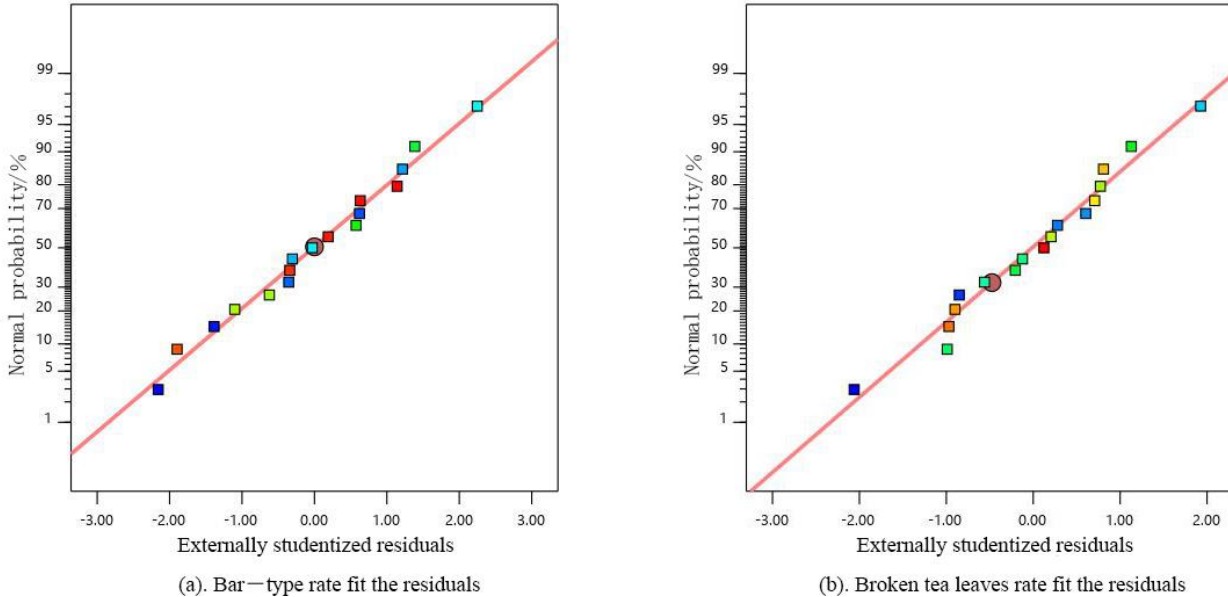

(a). Bar—type rate fit the residuals

(b). Broken tea leaves rate fit the residuals

**Figure 14.** Fit the residuals of bar-type rate and broken tea leaf rate.

*5.3. Response Surface Analysis*

The interrelationships between the factors and the evaluation indicators were analyzed using Design-Expert11 software to obtain 3D response surface plots of the test factors and their interactions. In the response surface model, second-order polynomial fits were used to express the true response parameters in the experimental design [28]. Based on regression equations, response surfaces were drawn, as shown in Figures 15 and 16. A combination of some influencing factors among convex bar height, pot slot angle of inclination, and slot number were fixed to analyze the effect of their interaction on the rate of bar-type and broken tea.

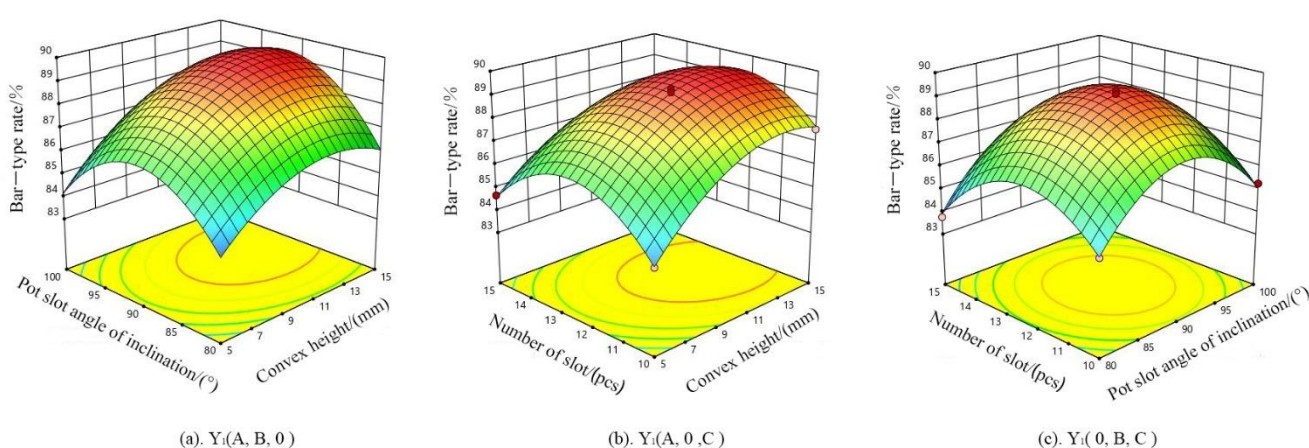

(a). $Y_1(A, B, 0)$

(b). $Y_1(A, 0, C)$

(c). $Y_1(0, B, C)$

**Figure 15.** Response surface analysis of influencing factors of bar-type rate.

As shown in Figure 15, an interaction was observed between the height of the convex bar, the angle of inclination of the pot slots, and the number of slots. These factors have a significant effect on the bar-type rate. At a constant height of the convex bar, the bar-type rate tends to increase and then decrease as the angle of inclination of the pot slot increases. When the pot slot inclination angle was certain, the bar-type rate increased and then decreased with the increasing height of the convex bar. When the height of the convex bar changed, a wide range of fluctuations was observed in the bar-type rate, and the effect of the convex bar height on the bar-type rate indicator was significant (Figure 15a). At a certain number of pot slots, the bar-type rate first increased and then decreased with the

increase in the height of the convex bar. At a certain height of the convex bar, the bar-type rate increased and then decreased as the number of slots increased. When the height of the convex bar changed, a wide range of fluctuations was observed in the bar-type rate, and the effect of convex bar height on the bar-type rate indicator was significant (Figure 15b). At a certain angle of inclination of the pot slot, the bar-type rate first increased and then decreased with the increase in the number of slots. At a certain number of slots, the bar-type rate increased and then decreased as the angle of inclination of the pot slot increased. When the angle of inclination of the pot slot changed, a wide range of fluctuations in bar-type rates was observed, and the effect of the pot slot angle of inclination on the bar-type rate indicator was significant (in Figure 15c). The indexes that affect the size of the bar-type rate from high to low are, in the following order: the convex bar height, the slot tilt angle, and the number of slots.

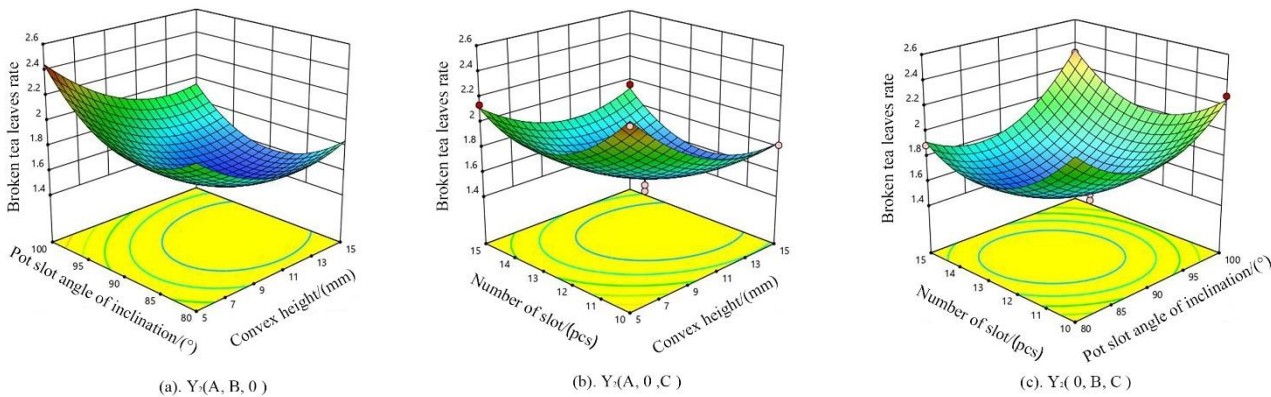

**Figure 16.** Response surface analysis of influencing factors of broken tea leaf rate.

As shown in Figure 16, an interaction was observed between the height of the convex bar, the angle of inclination of the pot slots, and the number of slots, and it remarkably affected the broken tea leaf rate. When the height of the convex bar was determined, the percentage of broken tea decreased and then increased as the inclination angle of the pot slot increased. When the angle of inclination of the pot slot was determined, the percentage of broken tea first decreased and then increased with the increase in convex bar height. The fluctuation range of the broken tea rate was larger when the convex bar height changed. The effect of changing the height of the convex bar on the broken tea rate indicator was significant (Figure 16a). At a certain number of slots, the percentage of broken tea decreased with the increasing height of the convex bars. When the convex bar height was constant, the percentage of broken tea decreased and then increased as the number of slots increased. Large fluctuations were observed in the rate of broken tea leaves when the height of the convex bar changed. The effect of the convex bar height on the broken tea rate index was significant (Figure 16b). When the angle of inclination of the pot slot was determined, the percentage of broken tea decreased and then increased as the number of slots increased. When the number of slots was determined, the percentage of broken tea decreased and then increased as the inclination angle of the pot slot increased. The fluctuation range of the broken tea leaves rate increased when the pot slot angle of inclination changed. The influence of the inclination angle of the pot slot on the broken tea rate index was significant (Figure 16c). The indicators that affect the size of the broken tea leaves rate are as follows: height of the convex bars > angle of inclination of the pot slot > number of slots.

In order to obtain the optimal parameter group of the performance of the tea de-enzyming and carding machine combined with the boundary conditions of the experimental factors, the optimal mathematical model of the tea de-enzyming and carding machine was established, and the regression equations of the bar-type rate and the broken rate of tea were analyzed. The parameter optimization module in Design-Expert software was used to solve the mathematical optimization model [29], and the optimization results were

obtained. When the convex bar height is 10.5975mm, the sloped angle of the groove is 91.1316°, and the number of the slots is 12.2983, the bar-type rate index is 89.6747%, the broken tea leaves rate index is 1.54253%, and the quality of the finished tea is the best.

### 5.4. Conclusion Verification

The study aimed to determine the uncertainty of the test, the repeatability of the test results, and the prediction of the regression model of the bar-type rate and the broken tea leaf rate of the tea by the de-enzyming and carding machine. Six groups of validation tests were carried out, and the test site is shown in Figure 17. The comparison of the regression model results for the percentage of tea bar-type and broken tea with the experimental results is shown in Table 7. The 6CSL-800 tea de-enzyming and carding machine was used as the verification test prototype. The maximum value of the bar-type rate index and the minimum value of the broken tea rate index were obtained. Analysis was carried out using Design-Expert 11 software to obtain the optimum combination of parameters for the de-enzyming and carding machine. The height of the convex bar was 10 mm, the angle of inclination of the pot slot was 90°, the number of slots was 12, and three groups (1, 2, 3) of control tests were carried out on the test prototype with the combination of these three parameters. A proper amount of fresh tea leaves was added and subjected to carding for 10 min. The actual values of the bar-type rate of the de-enzyming and carding machine obtained from the three control tests were 88.19%, 90.37%, and 87.33%, with an average value of 88.63%. The relative errors of their predicted and actual values are 1.43%, 1.02%, and 2.43%, with an average relative error of 1.63%. The actual values of the broken tea leaf rate obtained from the three sets of control tests for the de-enzyming and carding machines were 1.66%, 1.69%, and 1.61%, with an average value of 1.65%. The relative errors of their predicted and actual values were 1.81%, 3.56%, and 1.24, with an average relative error of 2.23%. The maximum error of the six controlled trials was 3.56%, with an average relative error of 1.92%. The validation results of the experiment are in general agreement with the parameter optimization results. The regression model proved to be accurate. Other conditions remain unchanged. Three control tests (4, 5, and 6) were carried out by changing the height of the convex bars to 15 mm. A proper amount of fresh tea leaves was added and subjected to carding for 10 min. The actual values of the bar-type rate of the de-enzyming and carding machine obtained from the three control tests were 86.28%, 85.89%, and 86.43%. The relative errors of their predicted and actual values were 1.47%, 1.93%, and 1.30%, with an average relative error of 1.57%. The actual values of the broken tea leaf rate obtained from the three sets of control tests for the de-enzyming and carding machines were 1.86%, 1.77%, and 1.89%. The relative errors of their predicted and actual values were 2.15%, 2.82%, and 3.71%, with an average relative error of 2.89%. The maximum error of the six controlled trials was 3.71%, with an average relative error of 2.23%. The validation results are in general agreement with the parameter optimization results, indicating that the regression model is accurate.

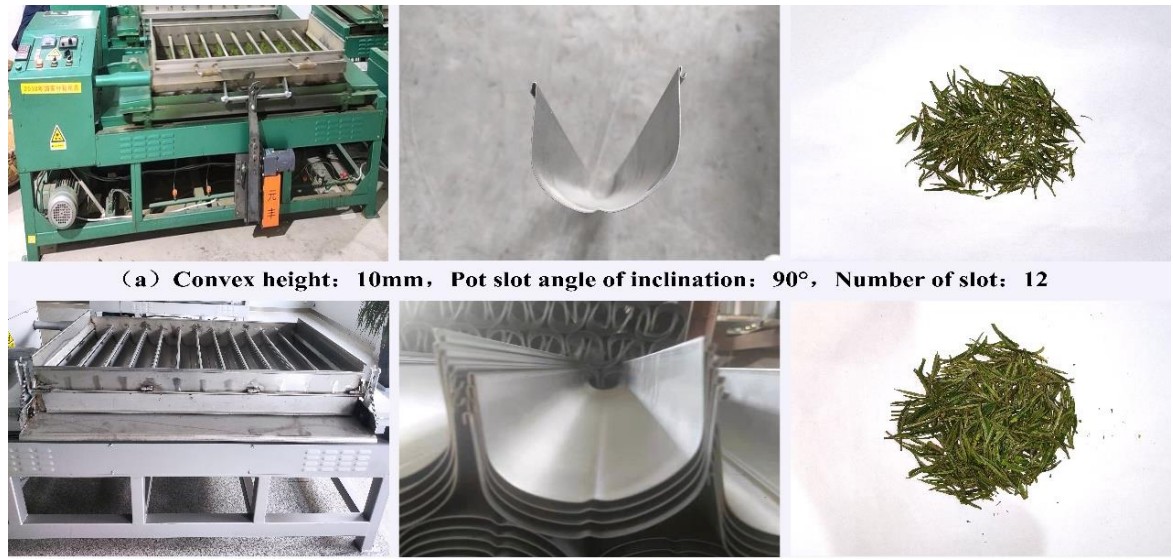

（a）Convex height：10mm，Pot slot angle of inclination：90°，Number of slot：12

（b）Convex height：15mm，Pot slot angle of inclination：90°，Number of slot：12

**Figure 17.** Test results of different control groups.

**Table 7.** Validation test results.

| Evaluation Index | Test Number | A | B | C | Predictive Value | Actual Value | Relative Error |
|---|---|---|---|---|---|---|---|
| $Y_1$ | 1 | 10 | 90 | 12 | 89.45 | 88.19 | 1.43 |
| | 2 | 10 | 90 | 12 | 89.45 | 90.37 | 1.02 |
| | 3 | 10 | 90 | 12 | 89.45 | 87.33 | 2.43 |
| | 4 | 15 | 90 | 12 | 87.55 | 86.28 | 1.47 |
| | 5 | 15 | 90 | 12 | 87.55 | 85.89 | 1.93 |
| | 6 | 15 | 90 | 12 | 87.55 | 86.43 | 1.30 |
| $Y_2$ | 1 | 10 | 90 | 12 | 1.63 | 1.66 | 1.81 |
| | 2 | 10 | 90 | 12 | 1.63 | 1.69 | 3.56 |
| | 3 | 10 | 90 | 12 | 1.63 | 1.61 | 1.24 |
| | 4 | 15 | 90 | 12 | 1.82 | 1.86 | 2.15 |
| | 5 | 15 | 90 | 12 | 1.82 | 1.77 | 2.82 |
| | 6 | 15 | 90 | 12 | 1.82 | 1.89 | 3.71 |

## 6. Conclusions

(1)   This study optimized and improved the pot slot structure of a tea de-enzyming and carding machine to improve the bar-type rate and reduce the rate of broken tea in the de-enzyming and carding machine by optimizing the height of the convex bar, the sloped angle of the slots and the number of the slots. Simulation modelling of tea particles was conducted using EDEM software. The working process for pot slots and pots was simulated with different convex bar heights, pot slot inclination angles, and numbers of slots, and the average speed and interaction force curves of tea particles were obtained. To further confirm the feasibility of the parameter optimization results, we carried out the scheme design according to the quadratic regression orthogonal combination rotation test. The results show that the sizes of the bar-type rate and the broken tea rate were affected by the parameters in the following order: convex bar heights > pot slot inclination angles > number of slots. Data were analyzed using Design-Expert software. The optimal combination of operating parameters obtained from the response surface optimization results is as follows: height of the convex bar, 10 mm; inclination angle of pot slot, 90°; and number of slots, 12. At this time, the bar-type rate was 89.45%, and the rate of broken tea was 1.63%. The validation test

yielded 88.63% bar-type rate, and the rate of broken tea was 1.65%. The results of the verification test are basically consistent with the results of parameter optimization.

(2) In this paper, the geometric model of the de-enzyming and carding machine prototype was generated using UG software, and the tea particle simulation model constructed using EDEM software can accurately analyze the movement of tea in the pot slot. The results can provide a theoretical reference for the design of subsequent tea de-enzyming and carding machines and similar tea machine equipment.

(3) Considering that the existing tea de-enzyming and carding equipment has been standardized, this paper takes the 6CSL-800 tea de-enzyming and carding machine as the research object. The effect of thermal deformation of the pots and uneven temperatures in the front and back of the pots caused by different heating methods on the efficiency of de-enzyming and carding was not considered. Further optimizations and improvements can be made in the subsequent design process to address these issues.

**Author Contributions:** Conceptualization, H.B., P.J., K.Q. and C.C.; methodology, H.B., P.J., K.Q. and L.Y.; software, H.B. and Y.B.; validation, H.B., C.C. and P.J.; formal analysis, P.J.; investigation, H.B.; data curation, H.B.; writing—original draft preparation, P.J.; writing—review and editing, H.B., P.J., K.Q. and C.C.; supervision, H.B. and L.Y.; funding acquisition, H.B. and K.Q. All authors have read and agreed to the published version of the manuscript.

**Funding:** This research was funded by the National Natural Science Foundation of China (grant number 52205509, 52105239), the Anhui Provincial Natural Science Foundation (grant number 2208085QE155), the Key Project of Natural Science Research in Anhui Universities (grant number KJ2021A0146), the Anhui Agricultural University Youth Fund (grant number k2135003) and by the Open Fund of State Key Laboratory of Tea Plant Biology and Utilization (grant number SKLTOF20220118), and by the Yuanfeng Tea Processing Technology Research and Quality Service Enhancement Enterprise Commissioning Project (grant number KJ2022436).

**Institutional Review Board Statement:** Not applicable.

**Informed Consent Statement:** Not applicable.

**Data Availability Statement:** Not applicable.

**Acknowledgments:** The authors would like to thank Xiaochun Wan from the State Key Laboratory of Tea Plant Biology and Utilization for providing equipment and technical support during the experiments. We also thank other colleagues in the laboratory, such as Kuan Qin, Chengmao Cao, and Yuxuan Bai, for their help in the experiments.

**Conflicts of Interest:** The authors declare no conflict of interest.

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
