# Peer review of "Optimization Design of Pot Slot Structure of Tea De-enzyming and Carding Machine"

_agronomy, doi:10.3390/agronomy12122937_

Round 1

Reviewer 1 Report

Papers presents interesting approach dealling with the determination of the movement trajectory and force of tea particles in the pot slot of a de-enzyming and carding machine in terms of the height of the convex bar in the pot slot, the inclination angle of the pot slot, and the number of slots. The paper is of theoretical character focused on the formulating mathematical model and solving optimization problem. In such type of consideration, in addition to general consideration, Authors should clearly state: 1) optimization problem, 2) objective function, and 3) mathematical model for optimization problem. Moreover, Authors should indicate why the problem under consideration is so imortant.

Author Response

Dear reviewer:

Thank you for your decision and constructive comments on my manuscript. We have carefully considered the suggestion of Reviewer and make some changes. We have tried our best to improve and made some changes in the manuscript.

According to your suggestions for modification, the red part that has been revised according to your comments. Revision notes, point-to-point, are given as follows:

Papers presents interesting approach dealling with the determination of the movement trajectory and force of tea particles in the pot slot of a de-enzyming and carding machine in terms of the height of the convex bar in the pot slot, the inclination angle of the pot slot, and the number of slots. The paper is of theoretical character focused on the formulating mathematical model and solving optimization problem. In such type of consideration, in addition to general consideration, Authors should clearly state: 1) optimization problem, 2) objective function, and 3) mathematical model for optimization problem. Moreover, Authors should indicate why the problem under consideration is so important.

  • optimization problem,

Response:

Thank you for your comments. In this study, the pot and groove structure of the tea de-enzyming and carding machine was optimized and improved. The structure of the pot groove is optimized mainly from the aspects of the height of the convex bar in the groove, the tilt Angle of the groove and the number of the groove. The relevant content has been added in the conclusion. (From line 740 to line 743).

  • objective function,

Response:

Thank you for your comments. The optimized objective function of this study is the interaction force between tea particles and the average velocity of tea particles, which is added in line 20-21 of the abstract.

  • mathematical model for optimization problem.

Response:

Thank you for your comments. According to your suggestions, we have modified the mathematical model of the optimization problem on the original basis to make it more vivid, image and specific. The optimized mathematical model is shown in FIG. 2, FIG. 3 and FIG. 4. (From line 144 to line 145 and 244 to 245 and 299 to 300 and 343 to 344).

Moreover, Authors should indicate why the problem under consideration is so important.

Response:

Thank you for your comments. Due to the lack of research on the optimization of pot and groove structure of tea de-enzyming and carding machine on the basis of existing research, and the different groove structure is directly related to the efficiency of tea de-enzyming and carding, it is necessary to consider these aspects to carry out research. The relevant content has been added in the introduction. (From line 91 to line 95).

Reviewer 2 Report

We recommend that you make the following changes to your manuscript

1. The keywords should be formulated in a more specific way, e. g. a certain analysis method was used or a certain type of simulation was used.

2. The introduction should consider adding a few more references to cite to articulate the background and purpose of the study more.

3. Some analysis of the results is missing from the manuscript and some discussion of the results should be properly added after the response surface analysis.

4. The content of the manuscript is too lengthy and appropriate deletion of repetitive expressions and content should be considered where appropriate.

5.The total number of references is relatively small and should be appropriately supplemented to make the text fuller. References preferably cited in the last five years, which reflect the latest research developments in the field.We consider your paper to be accepted with minor amendments.

Author Response

Dear reviewer:

We gratefully thanks for the precious time the reviewer spent making constructive remarks. We have tried our best to improve and made some changes in the manuscript. The red part that has been revised according to your comments. Revision notes, point-to-point, are given as follows:

  1. The keywords should be formulated in a more specific way, e. g. a certain analysis method was used or a certain type of simulation was used.

Response:

Thank you for your comments. We have perfected some keywords to make them more specific. EDEM is changed to discrete element simulation, and parameter optimization is changed to optimization of structural parameters. From line 42 to line 43.

  1. The introduction should consider adding a few more references to cite to articulate the background and purpose of the study more.

Response:

Thank you for your comments. In the introduction part, we have added relevant references in line with the research background of this paper, so as to enrich the research background and purpose of this paper. The relevant content has been added in the introduction. (From line 46 to line 47 and 47 to 51 and 85 to 90)

  1. Some analysis of the results is missing from the manuscript and some discussion of the results should be properly added after the response surface analysis.

Response:

Thank you for your comments. According to your suggestion, we added the discussion about the relevant results after response surface optimization in the last paragraph of section 5.3. (From lines 688 to 696).

  1. The content of the manuscript is too lengthy and appropriate deletion of repetitive expressions and content should be considered where appropriate.

Response:

Thank you for your comments. According to your suggestion, we have deleted some repeated expressions and contents in the article. However, we have retained some parts, because they are an important part of the article. If they are deleted, they cannot be fully expressed, resulting in the lack of logic.

  1. The total number of references is relatively small and should be appropriately supplemented to make the text fuller. References preferably cited in the last five years, which reflect the latest research developments in the field.

Response:

Thank you for your comments. According to your suggestions, we have added some references related to the research content of this paper, which are all from the last five years and represent the latest research achievements in this field. These references are added to lines 433-435, 561-562, and 692-693 of the article.

Reviewer 3 Report

Section 2.2 measuring principle gives a lengthy description of forces analysis and many equations are deduced. How are the analysis and simulation related and how are these equations used in the simulation should be explained more clearly in the text.

2.2.3 gives a force analysis of the collision between the tea particles and the inner wall of the pot slot, how about the collision between tea particles?

The number of slots should not be an experimental factor, as machines of different sizes definitely have number of slots, instead, the structural parameters of each slot are important, slot width, depth, bottom curvature etc. and should be optimized.

Introduction Ln 60-61, “foreign countries”, “domestic”, these expressions are inappropriate, since your paper is aimed at an international journal.

In text citations such as “by Su Zhongqiang [3] et al” are incorrect.

Abstract “The actual values of the bar-type rate obtained from the six sets of control tests 36 were 88.19%, 90.37%, and 87.33%, and the actual values of the broken tea rate were 1.66%, 1.69%, 37 and 1.61%, with average values of 88.63% and 1.65%.” This expression is confusing. Six sets of tests should have six results, you should be consistent with your expression, “three” or six”, how many tests, and how many groups?

Acknowledgements are written to thank people who provided support for this work but are not listed as authors of the paper. It is not appropriate to thank yourselves in the acknowledgements.

Author Response

Dear reviewer:

We gratefully appreciate for your valuable comments. We have tried our best to improve and made some changes in the manuscript. The red part that has been revised according to your comments. Revision notes, point-to-point, are given as follows:

Section 2.2 measuring principle gives a lengthy description of forces analysis and many equations are deduced. How are the analysis and simulation related and how are these equations used in the simulation should be explained more clearly in the text.

Response:

Thank you for your comments. In this paper, the force model of tea movement mainly analyzes the forces and various velocities, while the discrete element simulation in the third part analyzes the average velocity and interaction force of tea particles, and the curve diagram of the average velocity and interaction force of tea particles is also elaborated in detail in the simulation results analysis module in the fourth part of the paper. The specific content is in lines 377-437 and 438-548.

2.2.3 gives a force analysis of the collision between the tea particles and the inner wall of the pot slot, how about the collision between tea particles?

Response:

Thank you for your comments. According to your suggestion, the force analysis of the collision between tea particles has been added in Part 2.2.4 of the paper (From lines 335 to 376).

The number of slots should not be an experimental factor, as machines of different sizes definitely have number of slots, instead, the structural parameters of each slot are important, slot width, depth, bottom curvature etc. and should be optimized.

Response:

Thank you for your comments. Your suggestion is very reasonable, but it is necessary to explain to you here that there are 10 slot POTS, 12 slot POTS and 15 slot POTS in the tea de-enzyming and carding machine studied in this paper. Different types of slot POTS are a crucial factor affecting the efficiency of tea de-enzyming and carding. As for the factors such as the width, depth and bottom curvature of the pot groove you mentioned, we have not carried out research on them, and we will carry out research on them in further work.

Introduction Ln 60-61, “foreign countries”, “domestic”, these expressions are inappropriate, since your paper is aimed at an international journal.

Response:

Thank you for your comments. According to your suggestion, we have modified the relevant content, which is specifically modified in lines 65 to 67 of the article.

In text citations such as “by Su Zhongqiang [3] et al” are incorrect.

Response:

Thank you for your comments. According to your suggestions, we have made corresponding modifications in line 69 of the article.

Abstract “The actual values of the bar-type rate obtained from the six sets of control tests 36 were 88.19%, 90.37%, and 87.33%, and the actual values of the broken tea rate were 1.66%, 1.69%, 37 and 1.61%, with average values of 88.63% and 1.65%.” This expression is confusing. Six sets of tests should have six results, you should be consistent with your expression, “three” or six”, how many tests, and how many groups?

Response:

Thank you for your comments. The six groups of controlled trials described here are respectively the trials with a convex bar height of 10mm (Groups 1, 2 and 3) and the trials with a convex bar height of 15mm (Groups 4, 5 and 6) with other conditions unchanged. According to your suggestions, we have modified the relevant content in line 36-38 of the abstract.

Acknowledgements are written to thank people who provided support for this work but are not listed as authors of the paper. It is not appropriate to thank yourselves in the acknowledgements.

Response:

Thank you for your comments. According to your suggestions, we have revised relevant contents, specifically in lines 785 to 787 of the paper.
